# The misperception of Asian subgroup representation in STEM
A. Chyei Vinluan [1] ✉ & Michael W. Kraus [1,2]

The stereotype that Asian Americans excel in science and math has contributed to the narrative that they are overrepresented in STEM fields. However, U.S. Census data reveals this is not the case—there are significant disparities in STEM representation across Asian subgroups. The present research investigates whether U.S. participants are aware of these disparities. In Studies 1 and 2, we show that participants misperceive the STEM representation of Chinese, Japanese, Korean, Indian, Filipino, and Vietnamese subgroups. Study 3 demonstrates that these misperceptions persist despite changes in question framing and measurement. Furthermore, our findings suggest that these misperceptions are due to stereotypical expectations: participants view East Asian subgroups as more representative of Asian Americans and therefore more likely to be overrepresented in STEM, while perceiving Southeast Asian subgroups as less representative and more likely to be underrepresented. In a final study, we find that informing egalitarian-minded participants about these disparities increases support for racial equity–enhancing policies, and all participants' support for disaggregated data about Asian subgroups. Overall, our findings indicate that many U.S. participants are unaware of the within-group disparities among Asian Americans and underscore the importance of collecting and reporting data at the subgroup level to bring these inequalities to light.

In 1987, Time Magazine published an article about Asian American whiz kids excelling in science, engineering, and math[1]. The article captured a strong and enduring association between Asian Americans and the science, technology, engineering, and mathematics (STEM) field[2-5]. While many people consider this stereotypical association between Asian Americans and STEM as positive[6,7], there are costs associated with this stereotype. It can cause test anxiety among Asian American students[2,4,8,9] and can lead to inferences of Asian overrepresentation in STEM, which can create backlash[10]. In this research, we investigate how this stereotype is unevenly applied to Asian ethnic subgroups, including some while leaving others out of the STEM field.

Data on Asian Americans is often excluded from national U.S. surveys and rarely disaggregated by ethnic subgroups (e.g., Chinese, Japanese, Korean, Indian, Filipino, and Vietnamese)[11-13]. However, disaggregating by ethnic subgroup reveals within-group disparities, especially in the context of education—a domain that is often heavily associated with Asian American success. In 2020, only 33% of Vietnamese Americans were college graduates, compared with 60% of Chinese Americans and 79% of Indian Americans[14]. Similarly, when looking at graduate degrees, only 10% of Filipino Americans and 10% of Vietnamese Americans had earned one, compared with 29% of Chinese Americans and 43% of Indian Americans[15]. Despite narratives and

stereotypes that Asian Americans are educated and do well in school, these data show variation, with greater representation among Chinese and Indian Americans, and less representation among Filipino and Vietnamese Americans. Given the lack of information about Asian subgroups, many people are unlikely to understand cultural, ethnic, and representational variation within the Asian American category. Moreover, lay perceptions regarding academic achievement, including those as they relate to the representation of Asian subgroups in STEM, are likely to be inaccurate. The above analysis supports our first hypothesis that people will hold inaccurate beliefs about Asian subgroup representation in STEM.

Beyond inaccuracy, perceptions of Asian representation in STEM are likely to be shaped by expectations that more typical Asian subgroups (e.g., East Asians) are more strongly associated with STEM fields than less typical Asian subgroups (e.g., South and Southeast Asians). Typicality in person perception is the extent to which an individual closely matches the proto-type, or mental representation, of a group[16]. We generally expect that U.S. perceptions of subgroup typicality within the broader Asian American category will predict perceptions of subgroup representation in STEM. Previous work suggests that group-based stereotypes are more readily applied to more typical category members than less typical members[17-19]. Given existing stereotypes about Asian Americans in STEM[2-5], we expect

[1]Northwestern University, Department of Psychology, Evanston, IL, USA. [2]Northwestern University, Institute for Policy Research, Evanston, IL, USA.
✉e-mail: aeroelay.vinluan@northwestern.edu

people to be more likely to view East Asian subgroups as excelling in STEM compared to South or Southeast Asian subgroups, since East Asians are perceived as more representative of the Asian American category[20,21]. Perceptions of Asian typicality may also be related to narratives that Asian Americans are overrepresented in STEM[22–24]. That is, typicality may mean that overrepresentation in STEM refers to East Asian subgroups in particular. Additionally, Asian Americans are stereotyped as a high-status racial group within the U.S.[25], and based on previous research, we would expect that East Asian subgroups are perceived as having a higher status than South or Southeast Asian subgroups[26,27], and that STEM occupations are perceived as high status[28]. Thus, people may assume that East Asian subgroups are of a higher status due to being more likely to work in a STEM occupation than South or Southeast Asian subgroups[26,28–30].

Together, our analysis indicates that people are likely to hold beliefs that East Asian subgroups tend to be found in STEM occupations due to heightened expectations of their typicality and status among Asian origin subgroups. Thus, for our second hypothesis, we expected East Asian subgroups to be perceived as having greater representation in STEM than their population share, whereas the reverse would be true of representation for South or Southeast Asian subgroups. We have chosen to focus on perceptions of STEM representation relative to perceptions of population share because we are interested in subgroup representation in STEM after accounting for participant beliefs about subgroup population sizes. These latter population perceptions are a product of beliefs related to immigration history, current trends in migration, community size, and other factors. Thus, as opposed to STEM representation estimates computed relative to actual subgroup population size, which our participants are likely unaware of, perceptions of subgroup population help us understand subgroup STEM representation estimates, independent of population size error estimates, because they account for idiosyncratic theories about immigration history, current migratory patterns, and community size. In addition to our prediction of discrepancies between STEM and population estimates, we also expected participant beliefs about subgroup typicality and status to positively correlate with estimates of STEM representation. We examined these associations as part of the test of our second hypothesis.

In the context of testing our first two hypotheses, we also wanted to explore alternative explanations for the perceptual patterns we observed in Asian subgroup STEM estimates. One such alternative explanation centers around informational sources of STEM perceptions—that is, people with experiences among Asian subgroups or with experiences in STEM fields might, due to their experiences within these groups, have a more accurate understanding of which Asian subgroups can be found in STEM fields. For example, people's estimates of racial inequality were more accurate when people had more diverse social networks[31]. Research also indicates that misperceptions of resource distribution occur because people reference their social networks (e.g., family, friends, coworkers, and acquaintances) to make judgments about how attributes or resources are distributed within a population[32,33]. Therefore, we might expect that Asian individuals and STEM workers could be just as inaccurate in their perceptions of Asian subgroup representation in STEM as those who do not identify as Asian or work in STEM. This would suggest that misperceptions in STEM occur regardless of who is in one's social network.

While we have taken steps to measure perceptions of Asian subgroup STEM participation, these perceptions can be shaped by errors related to math and other comparative contexts[34,35]. In related research on misperceptions, scholars have varied the survey questions and reference groups used for assessing perceptions of inequality and group representation, to rule out scale or other context artifacts as explanations of perceptual patterns[35]. We employ similar methods in the present research, varying survey question response options and reference group comparisons, to determine if STEM estimates show consistent patterns across methodology. With these study design choices, we hope to test alternative methodological interpretations of our findings.

Our argument, thus far, suggests that people will misperceive the representation of Asian subgroups in STEM and that these misperceptions are related to psychological processes of subgroup typicality and status. One solution to these perceptual errors is to use information to counter stereotypes about Asian subgroup representation in STEM. Previous research suggests that providing people with such counter-narratives, under specific conditions that mitigate racial threat, has improved accuracy in prior research[36,37]. Thus, for our third hypothesis, we predicted that when people are presented with data on actual Asian subgroup representation in STEM, it would raise awareness about Asian subgroup inequality in STEM and thereby increase policy support for affirmative action policies that broaden representation in STEM for underrepresented groups. Although counter-narrative information has shaped policy beliefs in the context of group inequality in past research[36,38], the evidence of the effectiveness of these narratives is mixed[39]. One challenge is that some people prefer to maintain status hierarchies (i.e., social dominance orientation)[40,41] and are less likely to change their attitudes about support for hierarchy-attenuating policies (e.g., affirmative action)[42,43]. Thus, we explored, as part of our third hypothesis, whether learning counter-narratives about Asian subgroup representation in STEM would only increase support for affirmative action among people low in these hierarchy preferences as measured by social dominance orientation. We additionally explored one final consequence of providing people with counter-narratives about Asian subgroup representation in STEM: That such information would show the value of disaggregating the Asian category —in terms of who is marginalized by broad conceptions of Asian overrepresentation in STEM—and thereby increase support for similar disaggregation for data on Asian people. Such perceptions of the importance of disaggregation would be consistent with how organizations and legislatures are starting to collect data about Asian populations in the U.S.[44,45].

In the present research, we were interested in perceptions of the representation of Asian individuals with advanced STEM degrees, the processes that might explain these perceptions, as well as the consequences of being inaccurate. Therefore, we obtained STEM data from the U.S. Census Bureau[46] using their 2023 Annual Social and Economic Supplement (ASEC), a comprehensive work experience information on the employment status, occupation, and industry of people over the age of 15 years. The ASEC is conducted every year and does not include observations for most U.S. counties. Instead, the design is meant to provide consistent national-level estimates. The dataset contains an individual's highest degree earned (i.e., Master's or Ph.D.), their current field of occupation, and, importantly, the ethnic subgroup to which Asian Americans belong. Consistent with the National Science Foundation's (NSF) definition of STEM, we include only the Science and Engineering occupations: (1) Computer & mathematical occupations, (2) Architecture & engineering occupations, and (3) Life, physical, and social science occupations. From this dataset, we found that 32% of Chinese Americans, 2% of Japanese Americans, 3% of Korean Americans, 50% of Indian Americans, 2% of Filipino Americans, 2% of Vietnamese Americans, and 9% of Asian individuals from other ethnic subgroups have an advanced STEM degree. These results demonstrate that there is an unequal representation of advanced STEM degree holders within the Asian American category.

In Studies 1–2, we tested our first two hypotheses that (1) people will be inaccurate in their estimates of Asian subgroup representation in STEM and (2) that perceptions of East Asian representation in STEM will overestimate perceptions of the overall East Asian population. We also examined associations with perceptions of subgroup typicality and status. In Study 3, we attempted to replicate our results from the previous two studies using a different estimation method to rule out the possibility that STEM misperceptions were due to measurement effects. Finally, in Study 4, we tested our third hypothesis that presenting participants with actual data on the unequal representation of Asian subgroups in STEM will increase support for equity-enhancing policy, data disaggregation, as well as the potential moderating role of social dominance orientation.

## Methods

The materials and methods for Studies 1-3 were reviewed and approved by the Institutional Review Board at Yale University, while the Institutional

Review Board at Northwestern University approved the materials and methods for Study 4. For each study, we aimed to recruit at least $n = 200$ participants per condition or unit of between-subjects analysis (e.g., participant race). Study 1 was a large, racially diverse sample; Study 2 was a sample of participants who indicated they had an advanced STEM degree; and Studies 3–4 were convenience samples. Detailed demographic information about the samples in each study is provided in Table 1.

In all four studies, participants were recruited through Prolific, an online recruitment platform, to complete a 10-minute study about their "perceptions" and were compensated $2.00. An initial introductory screen informed participants that the study concerned how "individual personality is related to various social judgments" and that participation involved estimating the number of people who worked in certain fields. Participants were informed that they could skip any questions that they did not want to complete, with no loss of compensation or penalty. Participants indicated their consent to participate in the study by clicking the arrow on their computer screen to advance to the next page.

Study 2 was pre-registered prior to data collection on March 4, 2024 (link: https://osf.io/p5qj6/?view_only=0de4def437a943a2a58318ed05f4fb81). Additionally, Study 4 was pre-registered on January 6, 2025 (link: https://osf.io/cjuam/?view_only=63f29e0de3fb40d6b9e9711bf6874e48). However, Studies 1 and 3 were not pre-registered.

### Studies 1–3
After indicating their consent to complete the survey, participants in Studies 1-3 were first informed that they were going to make population estimates for each Asian subgroup. Specifically, we asked participants to answer the following: "If you had a random sample of 100 Asian Americans who live in the United States, how many would be from each of the subgroup categories below: Chinese, Asian Indian, Japanese, Korean, Filipino, Vietnamese, and other Asian". We informed participants that their responses should equal 100 and that the "other Asian subgroup" includes Pakistani, Thai, Cambodian, Hmong, Taiwanese, etc. Next, in Studies 1-2, participants completed the STEM estimation items for each subgroup and were given the following: "Within the United States, Asian Americans make up 34.7% of the advanced degrees (e.g., MA, MS, Ph.D.) in the Science, Technology, Engineering, and Mathematics (STEM) field. If you had a random sample of 100 Asian Americans with advanced STEM degrees, how many would be from each of the subgroup categories below: Chinese, Asian Indian, Japanese, Korean, Filipino, Vietnamese, and other Asian." We again told participants to enter their responses and that their responses should equal 100. Participants in Studies 1 and 2 were excluded from the final analyses if their total STEM estimates equaled less than 90 or greater than 110.

Participants in Study 3 completed a different version of the STEM estimation question and were randomly assigned to one of two conditions: the open-ended condition or the closed-ended condition. In the open-ended condition, participants read the following: "Within the United States, a career in the Science, Technology, Engineering, and Mathematics (STEM) fields often requires an advanced degree (e.g., MA, MS, Ph.D.). If you had a random sample of 100 Americans with advanced STEM degrees, how many would be from each of the categories below: White, Black, Chinese, Indian, Japanese, Korean, Filipino, and Vietnamese?" We asked participants to provide STEM estimates for the Black and White racial groups to help participants think about the entire U.S. population rather than just the U.S. Asian American population. Participants were asked to enter their responses. In the closed-ended condition, participants read the following: "Within the United States, a career in the Science, Technology, Engineering, and Mathematics (STEM) fields often requires an advanced degree (e.g., MA, MS, Ph.D.). What percentage of Americans from the following categories below have an advanced STEM degree: White, Black, Chinese, Indian, Japanese, Korean, Filipino, and Vietnamese?" Participants were presented with 12 options and asked to select their response: "0–0.9%", "1–1.9%", "2–2.9%", "3–3.9%", "4–4.9%", "5–5.9%", "6–6.9%", "7–7.9%", "8–8.9%", "9–9.9%", "10%", and "greater than 10%".

Afterward, participants in all three studies were asked to rate how typical they considered the six Asian subgroups of the Asian American group using a seven-point Likert scale (1 = not at all typical, 7 = very typical). Finally, participants indicated the status of the six Asian subgroups using the MacArthur Scale of Subjective Status[47] on a ten-point Likert scale (1 = worst off, 10 = best off). See Supplement Note 1 for typicality and status ANOVA results.

Following these survey responses, participants completed several additional questions about their beliefs about society, including social dominance orientation and perceived symbolic and realistic threat from Asian immigrants. The full list of questions for all studies is available online. Following these items, participants reported their demographic information (e.g., age, gender, race, educational attainment) and indicated how socially and economically conservative they considered themselves to be using a seven-point Likert scale (1 = very liberal, 7 = very conservative). Data distribution was assumed to be normal, but this was not formally tested.

### Study 4
After indicating their consent to complete the survey, participants in Study 4 were first informed that they were going to review government policies. However, before reviewing the policies, participants were informed they would be watching a video. Participants were randomly assigned to one of two video conditions: control or intervention. In the informational intervention condition, participants watched a 2:35-minute video where the narrator first introduced what fields are considered part of STEM. Next, the narrator discussed racial groups that are underrepresented in STEM, which include Filipino and Vietnamese Americans, but many people are unaware of this due to the overall Asian American category being represented in STEM. In the control condition, participants watched a 2:22-minute video where the narrator first introduced what fields are considered part of STEM, but then transitioned into facts about the STEM workforce and why people may consider a career in STEM. Importantly, in the control condition, there was no discussion about representation in STEM.

After, participants completed three items assessing support for affirmative action: "In general, do you think affirmative action programs in hiring, promoting, and college admissions should be continued, or do you think these affirmative action programs should be abolished?", "In general, do you think affirmative action programs in STEM degree programs should be continued, or do you think these affirmative action programs should be abolished?" (both measured on 1 *definitely should be abolished* - 4 *definitely should be continued* scales), and "In general, do you think affirmative action programs in STEM should include Vietnamese and Filipino students, or do you think these affirmative action programs should not include these students?" (measured on a 1 *definitely should not be included* - 4 *definitely should be included* scale). We also measured participants' support for the government collecting and presenting Asian American data at the ethnic subgroup level on a 1 (*strongly oppose*) - 4 (*strongly support*) scale.

Following these survey responses, participants answered eight items about their social dominance orientation[40] (SDO; e.g., "An ideal society requires some groups to be on top and others to be on the bottom"). Their responses were averaged to create a composite score.

Afterward, participants completed several additional questions about their perceptions of Asian Americans, including a feeling thermometer and the internalization of the model minority myth. The full list of questions for all studies is available online. Following these items, participants reported their demographic information.

### Reporting summary
Further information on research design is available in the Nature Portfolio Reporting Summary linked to this article.

## Results
### Accuracy of STEM representation perceptions
We first examined whether people knew of the unequal representation of Asian ethnic subgroups in STEM. We ran a series of one-sample t-tests for

**Table 1 | Participant demographic characteristics from studies 1–4**

| | Study 1 | Study 2 | Study 3 | Study 4 |
|---|---|---|---|---|
| Sample Size, N | 784 | 197 | 451 | 708 |
| Exclusions, n | 15 | 32 | | |
| Conditions, n | | | | |
| Open-ended | | | 226 | |
| Close-ended | | | 227 | |
| Control | | | | 354 |
| Intervention | | | | 354 |
| Age, mean (SD) | 37.72 (12.70) | 43.97 (13.50) | 37.88 (11.69) | 46.54 (25.07) |
| Gender Identity, n | | | | |
| Man | 369 | 106 | 189 | 341 |
| Woman | 383 | 86 | 255 | 354 |
| Non-binary | 14 | 5 | 7 | 9 |
| Racial Identity, n | | | | |
| White | 190 | 118 | 338 | 439 |
| Black | 180 | 37 | 25 | 103 |
| Asian | 192 | 23 | 43 | 43 |
| East, South, Southeast | 107, 22, 63 | 10, 8, 5 | 16, 10, 17 | 21, 4, 18 |
| Latinx | 134 | 5 | 19 | 45 |
| Pacific Islander | 71 | 12 | 25 | 48 |
| Education, n | | | | |
| Some High School | 2 | | 4 | 6 |
| High School/GED | 79 | | 52 | 75 |
| Some College | 131 | | 93 | 130 |
| Associate's | 72 | | 41 | 81 |
| Bachelor's | 335 | | 165 | 271 |
| Some Graduate School | 23 | | 8 | 13 |
| Master's | 116 | 164 | 63 | 113 |
| Doctoral | 10 | 33 | 9 | 15 |
| Occupational Field, n | | | | |
| Computer & Math Science | 90 | | 44 | 72 |
| Architecture & Engineering | 29 | | 7 | 14 |
| Life, Physical, & Social Science | 16 | | 7 | 15 |
| Non-STEM field | 768 | | 393 | 603 |
| STEM degree, n | | | | |
| Agriculture, Forestry, Fisheries, & Veterinary | | 6 | | |
| Engineering, Manufacturing, & Construction | | 38 | | |
| Information & Communication Technologies | | 48 | | |
| Math & Statistics | | 24 | | |
| Natural Sciences | | 33 | | |
| Social Sciences | | 47 | | |
| Conservatism, mean (SD) | 3.19 (1.65) | 3.36 (1.76) | 3.20 (1.67) | 3.86 (1.82) |

each subgroup comparing participants' average STEM estimates to the percentages provided by the U.S. Census Bureau[46]. Given that the method and measures of Studies 1-2 were the same, we report combined meta-means ($M_{meta}$) and meta-analytic effect size estimates (i.e., Fisher's $z$-transformed correlation coefficient; $z_{Fisher}$) in the main text[48] but see Table 2 for individual study results. We conducted the meta-analysis by first converting the Cohen's $d$ effect size from the one-sample t-tests into correlation coefficients ($r$) using the following equation: $r = \frac{d}{\sqrt{d^2+4}}$. Then, we used the *metafor* R package[49] to calculate the average effect size for each Asian subgroup.

Across two studies ($N = 981$), we found evidence that participants misperceived the representation of each Asian subgroup in STEM, supporting our first hypothesis that people will be inaccurate in their estimates of Asian subgroup representation in STEM. Participants estimated $M_{meta} = 26.16$, $95\%$ Confidence Interval ($CI$) [25.42, 26.91] individuals have an advanced STEM degree for the Chinese subgroup, which is less than the actual percentage for the Chinese subgroup (32%), $z_{Fisher} = -0.243$, $p < .0001$, $95\%CI$ [−0.31, −0.18]. Additionally, participants estimated $M_{meta} = 13.87$, $95\%CI$ [12.37, 15.38] for the Japanese subgroup and $M_{meta} = 11.94$, $95\%CI$ [11.53, 12.35] for the Korean subgroup, both of which are more than the actual percentage for the Japanese (2%), $z_{Fisher} = 0.679$, $p < .0001$, $95\%CI$ [0.62, 0.74], and Korean subgroups (3%), $z_{Fisher} = 0.645$, $p < .0001$, $95\%CI$ [0.58, 0.71]. Participants estimated $M_{meta} = 23.96$, $95\%CI$ [20.42, 27.50] for the Indian subgroup, which is less than the actual percentage (50%), $z_{Fisher} = -0.863$, $p < .0001$, $95\%CI$ [-1.00, -0.73]. Moreover, participants estimated $M_{meta} = 7.96$, $95\%CI$ [7.14, 8.79] for the Filipino subgroup, and $M_{meta} = 7.75$, $95\%$ $CI$ [7.39, 8.10] for the Vietnamese subgroups, both of which are more than the actual percentage for the Filipino (2%), $z_{Fisher} = 0.489$, $p < .0001$, $95\%CI$ [0.43, 0.55] and Vietnamese subgroups (2%), $z_{Fisher} = 0.485$, $p < .0001$, $95\%CI$ [0.42, 0.55]. Finally, participants estimated $M_{meta} = 8.74$, $95\%CI$ [8.21, 9.27] for the other subgroup, which did not significantly differ from the actual percentage (9%), $z_{Fisher} = -0.015$, $p = .638$, $95\%CI$ [−0.08, 0.05]. These findings suggest that U.S. participants are unaware of the number of individuals in STEM by Asian subgroup, providing evidence that participants do not know that there is variation in representation in STEM. In the next section, we test alternative explanations for participants' inaccuracies.

**Social network and measurement effects.** Our findings, thus far, indicate that U.S. participants are inaccurate when it comes to Asian subgroup representation in STEM. However, we wanted to begin to rule out other possible explanations for our findings. First, we checked the possibility that these inaccuracies are due to U.S. participants' social networks lacking either Asian people or STEM workers. We had expected that if this was the case, then Asian participants and participants with an advanced STEM degree might be more accurate. However, that was not the case. In Study 1, while Asian participants were slightly more accurate in estimating Indian and Japanese subgroup representation in STEM than other racial groups (see Supplementary Note 2 and Table S3), Asian participants were still inaccurate, just like the other racial groups (see Supplementary Note 2 and Table S4). In Study 2, participants with an advanced STEM degree were also still inaccurate in their estimations (see Table 1). Thus, it seems that these inaccuracies were not due to the lack of Asian people or STEM workers in participants' social networks.

Another source of participants' inaccuracies could be the context and method of measuring participants' perceptions. We addressed these possibilities in Study 3. Specifically, participants were asked about the STEM representation of each Asian subgroup relative to the entire U.S. population instead of relative to just the U.S. Asian American population, as in Studies 1-2. We also manipulated how participants made their STEM estimations by asking participants to either enter their responses (i.e., the open-ended condition) or select their responses from one of the provided categories (i.e., the closed-ended condition). Despite these changes in context and method,

**Table 2 | Descriptive statistics and t-test results for Study 1 (N = 784) and Study 2 (N = 197)**

| Subgroup | Study | STEM Estimates | | | Population Estimates | | | Paired Samples t-value |
|---|---|---|---|---|---|---|---|---|
| | | Mean (SD) | Actual | *t*-value | Mean (SD) | Actual | *t*-value | |
| Chinese | 1 | 26.19 (11.67) | 32 | −13.94* | 25.04 (10.79) | 23 | 5.29* | 3.09* |
| | 2 | 26.03 (12.84) | 32 | −6.53* | 25.22 (11.76) | 23 | 2.65 | 1.15 |
| Japanese | 1 | 14.57 (8.49) | 2 | 41.48* | 12.58 (7.09) | 5 | 29.96* | 7.40* |
| | 2 | 13.03 (7.89) | 2 | 19.62* | 11.81 (6.59) | 5 | 14.50* | 2.34 |
| Korean | 1 | 12.02 (6.49) | 3 | 38.89* | 12.07 (6.35) | 8 | 17.92* | −0.18 |
| | 2 | 11.63 (6.41) | 3 | 18.90* | 11.71 (6.00) | 8 | 8.69* | −0.22 |
| Indian | 1 | 22.25 (13.17) | 50 | −58.98* | 17.97 (10.64) | 33 | −39.58* | 10.85* |
| | 2 | 25.87 (14.00) | 50 | −24.19* | 21.47 (11.23) | 33 | −14.41* | 6.10* |
| Filipino | 1 | 8.34 (6.35) | 2 | 27.97* | 12.39 (7.16) | 12 | 1.53 | −15.63* |
| | 2 | 7.49 (5.01) | 2 | 15.39* | 11.49 (6.41) | 12 | −1.12 | −9.66* |
| Vietnamese | 1 | 7.81 (5.70) | 2 | 28.54* | 10.48 (6.72) | 6 | 18.66* | −10.91* |
| | 2 | 7.50 (5.69) | 2 | 13.55* | 9.44 (5.91) | 6 | 8.18* | −4.85* |
| Other | 1 | 8.77 (8.64) | 9 | −0.74 | 9.85 (9.08) | 14 | −12.80* | −3.65* |
| | 2 | 8.63 (7.84) | 9 | −0.66 | 8.85 (8.51) | 14 | −8.48* | −0.42 |

Descriptive statistics, individual one-sample t-test results comparing participants' STEM and population estimates to actual STEM and population percentages provided by the U.S. Census, and paired sample t-test results comparing participants' STEM to their population estimates for Study 1 (N = 784) and Study 2 (N = 197) by Asian subgroup. *p < .0071 – p-value adjusted for a Bonferroni correction (7 comparisons).

we replicated our previous finding that participants' estimations of Asian subgroup representation in STEM were inaccurate (see Supplementary Note 3). While the changes in the context did result in participants underestimating Chinese STEM representation in the open-ended condition, we still found the same pattern of estimation errors for Japanese, Korean, Indian, Filipino, and Vietnamese subgroup STEM representation, despite having more conservative percentage options in the closed-ended condition. Thus, our Study 3 results suggest that participants' inaccuracies are not due to the context and method of measurement.

Participants also misestimated the representation of White and Black Americans in STEM. In the open-ended condition, participants underestimated the representation of White Americans in STEM and overestimated the representation of Black Americans in STEM. In the closed-ended condition, participants underestimated Black representation in STEM. See Supplementary Note 3.

**Perceptions of over- or underrepresentation in STEM**

In the previous analyses, we found evidence that U.S. participants were inaccurate about the representation of Asian subgroups in STEM and provided data that called into question explanations of these inaccuracies arising from social network contact or measurement artifacts. In the next set of analyses, we examined whether participants' perceptions of STEM representation were due to stereotypic expectations that more typical Asian subgroups (e.g., Chinese, Japanese, Korean) are more strongly associated with STEM fields than less typical Asian subgroups (e.g., Indian, Filipino, Vietnamese). We conducted paired samples t-tests comparing participants' STEM estimates to their population estimates for each subgroup (Table 2), but report combined meta-means ($M_{meta}$) and meta-analytic effect size estimates (i.e., standardized mean difference; $d$) in the main text. We conducted the meta-analysis by using Cohen's $d$ obtained from the paired samples t-test for each study. Then, we used the *metafor* R package[47] to calculate the average effect size for each Asian subgroup.

Unlike comparisons between STEM estimates and real population size, comparison of STEM and population estimates allows us to determine whether participants think specific Asian ethnic subgroups are over- or under-represented in STEM, according to their own beliefs about population size. This analysis allows us to measure STEM representation while accounting for beliefs that participants have about subgroup immigration history, current migratory patterns, and community size. If participants'

STEM estimates were greater than their population estimates for a subgroup (i.e., participants perceive the subgroup to be overrepresented in STEM), we could infer that the subgroup was seen as well represented in STEM, relative to the perceived population. However, if participants' STEM estimates were smaller than their population estimates for a subgroup (i.e., participants perceive the subgroup to be underrepresented in STEM), this indicates the group is not well-represented in STEM.

Consistent with our second hypothesis, participants' STEM estimates for the Chinese subgroup ($M_{meta}$ = 26.16, 95%CI [25.42, 26.91]) were greater than their population estimates ($M_{meta}$ = 25.07, 95%CI [24.38, 25.76]), $d$ = 0.104, $p$ = .0011, 95%CI [0.04, 0.17]. Similarly, participants' STEM estimates for the Japanese subgroup ($M_{meta}$ = 13.87, 95%CI [12.37, 15.38]) were greater than their population estimates ($M_{meta}$ = 12.30, 95%CI [11.58, 13.03]), $d$ = 0.235, $p$ < .0001, 95%CI [0.15, 0.32]. These results suggest that participants consider the Chinese and Japanese subgroups to be overrepresented in STEM. Interestingly, participants' STEM estimates for the Korean subgroup ($M_{meta}$ = 11.94, 95%CI [11.53, 12.35]) did not significantly differ from their population estimates ($M_{meta}$ = 11.99, 95%CI [11.60, 12.38]), $d$ = −0.008, $p$ = .7922, 95%CI [−0.07, 0.05]. For two of the three East Asian subgroups, our participants thought these groups were well-represented in STEM relative to their share of the population. See Fig. 1.

For the Southeast Asian subgroups, perceptions tended to reflect a sense of a lack of representation for Filipino and Vietnamese subgroups. Participants' STEM estimates for the Filipino subgroup ($M_{meta}$ = 7.96, 95%CI [7.14, 8.79]) were fewer than their population estimates ($M_{meta}$ = 12.02, 95%CI [11.15, 12.89]), $d$ = −0.603, $p$ < .0001, 95%CI [−0.72, −0.48]. Similarly, participants' STEM estimates for the Vietnamese subgroup ($M_{meta}$ = 7.75, 95%CI [7.39, 8.10]) were fewer than their population estimates ($M_{meta}$ = 10.02, 95%CI [9.01, 11.03]), $d$ = -0.380, $p$ < .0001, 95%CI [−0.44, −0.32] – participants consider the Filipino and Vietnamese subgroups to be underrepresented in STEM. Finally, participants' STEM estimates for the other subgroup ($M_{meta}$ = 8.74, 95%CI [8.21, 9.27]) were fewer than their population estimates ($M_{meta}$ = 9.48, 95%CI [8.54, 10.42]), $d$ = -0.099, $p$ = .0334, 95%CI [−0.19, −0.01].

Interestingly, participants' STEM estimates for the Indian subgroup ($M_{meta}$ = 23.96, 95%CI [20.42, 27.50]) were greater than their population estimates ($M_{meta}$ = 19.65, 95%CI [16.22, 23.07]), $d$ = 0.396, $p$ < .0001, 95%CI [0.33, 0.46]. Perhaps this pattern reflects unique insights about Indian representation in STEM fields, an insight we return to in the discussion.

**Fig. 1 | STEM and population estimates by Asian ethnic subgroup for Study 1 (N = 784) and Study 2 (N = 197).** Estimates are separated by study – Study 2 estimates are a lighter shade than Study 1 estimates. Circles represent STEM while squares represent population estimates. Each Asian subgroup is represented by a distinct color. Each point represents a participant's response – clustered points indicate more common responses. The black squares with brackets indicate means and 95% confidence intervals surrounding the mean.

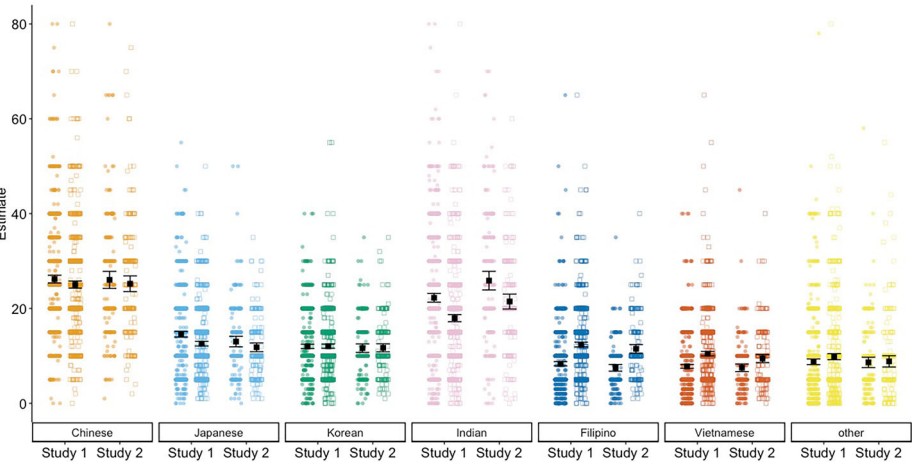

Overall, these analyses provide some evidence that two East Asian subgroups (e.g., Chinese, Japanese) are seen as more representative of STEM than their population share would indicate, whereas the two Southeast Asian subgroups (e.g., Vietnamese, Filipino) were seen as not well-represented based on their perceived population share. These patterns of findings could be due to a number of psychological and structural factors. To begin to better understand these patterns, we examined participants' perceptions of subgroup typicality and status.

**Perceived Asian typicality and status**. We found evidence that participants' STEM perceptions tracked stereotypic expectations of which Asian subgroups would be represented in STEM, but these analyses could reflect at least two distinct stereotypic expectations: that more typical Asian subgroups would be associated with greater representation in the STEM category, or that perceptions of Asian subgroup status would be associated with STEM representation[25,26]. Thus, we hypothesized that people's perceptions of Asian typicality and the status of an Asian subgroup would predict STEM representation perceptions for that subgroup.

We conducted a series of separate multi-level models (MLMs) using the *lmer* R package[50] with perceived Asian typicality or perceived status as the predictor variable, Asian subgroup as a moderator (reference = Chinese), and the difference between participants' STEM estimate - population estimate as the dependent variable. Positive numbers for this difference score indicate the overrepresentation of an Asian subgroup, while negative numbers indicate underrepresentation. We conducted each model without and with demographic control variables, and the results remain largely consistent across these models (Tables S7 and S9). We again meta-analyzed the regression results, given the similarity in methods and measures by calculating semipartial correlations ($r_{sp}$)[51]. We report the results for the models with demographic control variables in the main text. We would have evidence for our predictions if we observed an overall positive association of typicality or status with overrepresentation in STEM.

Consistent with our expectations, the model examining typicality ratings found that perceiving a subgroup as more typical of the Asian American category was associated with perceptions of overrepresentation in STEM, $r_{sp} = 0.07$, *95%CI* [0.01, 0.14], $p = .023$. Interestingly, there was some significant variation in typicality rating associations with STEM estimates among the Filipino and Indian subgroups (Table S6).

Also consistent with our predications, higher status ratings for an Asian subgroup predicted greater perceptions of overrepresentation in STEM, $r_{sp} = 0.19$, *95%CI* [0.13, 0.27], $p = .007$. Interestingly, there was some significant variation in status rating associations with STEM estimates among the Filipino and Vietnamese subgroups (Table S8). These findings suggest that the stereotypic expectations driving participants' STEM misperceptions for each Asian subgroup may be based on participants' perceptions of Asian typicality and status for each subgroup.

**Disaggregated data intervention**

We hypothesized that presenting people with information to counter stereotypes about Asian subgroup representation in STEM would increase awareness about Asian subgroup inequality and, as a result, increase support for affirmative action to broaden representation in STEM for underrepresented groups and for the collection of disaggregated data at the Asian subgroup level. Therefore, in Study 4, we designed an intervention where we informed participants about the underrepresentation of Filipino and Vietnamese individuals in STEM and focused on the consequences for people not knowing that these two Asian subgroups are underrepresented in STEM.

The intervention did not change people's support for affirmative action in general (control: $M = 2.83$, $SD = 1.05$; intervention: $M = 2.79$, $SD = 1.08$), $t(706) = 0.43$, $p = .671$, $d = 0.032$, *95% CI* [−0.12, 0.18] or in STEM (control: $M = 2.94$, $SD = 1.07$; intervention: $M = 2.88$, $SD = 1.08$), $t(706) = 0.74$, $p = .463$, $d = 0.055$, *95% CI* [−0.09, 0.20]. Additionally, an independent sample t-test revealed that there was not an overall significant difference in participants' reported support for affirmative action to include Southeast Asian Americans between the intervention ($M = 3.28$, $SD = 0.92$) and control conditions ($M = 3.23$, $SD = 0.87$), $t(706) = 0.76$, $p = .449$, $d = 0.057$, *95% CI* [−0.20, 0.09].

As an exploratory analysis, we examined the interaction between intervention conditions and SDO[40], or support for social inequality and hierarchy, on affirmative action support to include Southeast Asian Americans—given that people who support the formation and maintenance of hierarchy may be unwilling to expand equitable policies regardless of messaging. Regression results show that SDO significantly and negatively predicted affirmative action support, $b = −0.20$, $p < .0001$, *95%CI*[−0.27, −0.13], and when SDO is included in the regression model, affirmative action support to include Southeast Asian Americans was significantly higher in the intervention than in the control condition, $b = 0.51$, $p = .0003$, *95%CI*[0.23, 0.78]. Furthermore, SDO significantly moderated the relationship between intervention condition and affirmative action support, $b = −0.17$, $p = .0005$, *95%CI* [−0.27, −0.08]. We followed up the significant interaction with a simple slopes analysis: for participants low in SDO (-1SD = 1.27), participants in the intervention condition reported greater affirmative action support to include Southeast Asian Americans than participants in the control condition, $b = 0.29$, $p < .0001$, *95%CI* [0.12, 0.46]. However, there was not a significant difference in affirmative action support for participants at the mean level of SDO ($M = 2.53$), $b = 0.07$, $p = .240$, *95% CI* [−0.05, 0.19], and those high in SDO (+1 SD = 3.79), $b = −0.14$, $p = .100$, *95%CI* [−0.32, 0.03]. The SDO moderation analysis replicates previous research findings that participants high in SDO are less likely to support hierarchy-attenuating policies like affirmative action[42,43]. Additionally, SDO did not moderate the relationship between condition and support for the general, $p = .199$, and STEM-specific affirmative action items, $p = .102$.

We additionally measured participants' support for the government collecting and presenting Asian American data at the ethnic subgroup level. Participants in the intervention condition ($M = 2.85$, $SD = 0.86$) reported greater support for the government collecting and presenting disaggregated Asian American data than participants in the control condition ($M = 2.57$, $SD = 0.90$), $t(706) = -4.30$, $p < .0001$, $d = -0.323$. Together, these results suggest that when participants learn about the underrepresentation of Filipino and Vietnamese Americans in STEM, participants are more likely to support affirmative action, but only those low in SDO (i.e., those who prefer social equality and dismantling hierarchies), and are more likely to support the government collecting and presenting disaggregated Asian American data.

## Discussion

The stereotypical association between Asian Americans and STEM fields in the U.S. is often viewed as positive, but the potential downsides of this association are frequently overlooked. Our research shows that one significant cost is the widespread misperception that Asian subgroups are equally represented in STEM. In particular, there is a general lack of awareness about the existing disparities among Asian subgroups in STEM representation: Participants were inaccurate in the number of individuals represented in STEM by Asian subgroup, which cannot be attributed solely to measurement methods or network homophily. Additionally, participants tend to perceive East and South Asian subgroups as overrepresented in STEM, while Southeast Asian subgroups are viewed as underrepresented. We find that these misperceptions reflect perceptions of over- and under-estimation based on perceived population among East Asian and South/Southeast Asian groups, respectively, and are related to participants' views of each subgroup's typicality and perceived social status. Overall, our findings suggest that people in the U.S. are unaware of disparities in the representation of Asian subgroups in STEM, and that these perceptions are shaped by the stereotypical belief that STEM is more strongly associated with East and South Asian subgroups than with Southeast Asian subgroups.

Interestingly, our findings suggest that perceived Asian typicality varies depending on context. We had expected that participants would perceive the Indian subgroup to be underrepresented in STEM given that Indian Americans are generally perceived as less typical of the Asian American category than both East and Southeast Asian subgroups[20,21]. However, our results indicate that participants consider Indian Americans to be over-represented in STEM rather than underrepresented, suggesting that within the STEM context, Indian Americans are perceived as more typical of the Asian American category. In fact, supplemental analyses of perceived Asian typicality ratings show that while the Indian subgroup was still rated as less typical of the Asian American category than East Asian subgroups, they were rated as typical of the Asian American category as Southeast Asian subgroups. This latter finding is inconsistent with previous research[20,21]. Thus, it seems likely that Indian Americans are perceived as more typical of the Asian American category in STEM-context compared to unspecified contexts. The strong association of Indian Americans and STEM could be specifically due to Indian Americans' strong representation in the tech industry. For example, the H-1B visa program awarded 73% of the visas to people from India, and the most common H-1B occupations were computer-related jobs[52]. Additionally, the CEOs of major technology companies such as Alphabet Inc. (Google's parent company), Microsoft, Adobe, and IBM are of South Asian descent[53]. Therefore, future research should consider the role of context in perceptions of Asian typicality.

Our work also suggests that a larger issue is that diverse social categories, like Asian Americans, are often susceptible to cognitive biases. Specifically, people tend to *default* to automatically thinking of the needs of typical category members rather than considering the needs of the entire category[54,55]. In the context of STEM, Asian Americans, or more specifically, East Asian Americans, are stereotyped to excel in STEM, leading people to think that all Asian subgroups excel in STEM (e.g., the model minority myth[4,6]). However, these assumptions lead our participants to be unaware of how underrepresented Filipino and Vietnamese Americans (i.e., less typical

category members) are in STEM. Our findings suggest that cognitive biases that East Asian Americans are the Asian default[20,21] may lead people to forget or ignore the diversity of the Asian American category.

Our findings also provide insight into a potential solution to address these inequalities in STEM representation. Specifically, informational interventions highlighting the underrepresentation of Southeast Asian subgroups in STEM can increase support for the government collecting and presenting disaggregated data (i.e., data at the Asian subgroup level) as well as support for affirmative action, but only for participants who report having a low social dominance orientation. We theorized that this was the case because participants learned that there were inequalities in STEM and contradicted their beliefs that all Asian Americans are overrepresented in STEM[10], leading to a desire to address these issues. If increasing awareness of the underrepresentation of Southeast Asian subgroups in STEM can lead to greater support for racial equity policies, then disaggregated data may also provide additional benefits for advancing the broader Asian American community. Most importantly, disaggregated data can highlight the unequal representation of Asian subgroups in STEM - particularly the underrepresentation of Southeast Asian subgroups. Only when the U.S. public and policymakers are aware of these disparities can resources be effectively allocated to improve representation.

Interestingly, Asian participants are just as likely to be inaccurate about subgroup representation in STEM as other racial groups. This finding runs counter to expectations that within-group knowledge might elicit some special understanding of representation[56,57]. Instead, perhaps Asian people in the U.S. are subject to the same stereotypes as other racial groups and thus exhibit the same expectations of STEM representation[8]. Future research could more definitively test these alternatives by sampling particular Asian subgroups who are more and less well-represented in STEM.

Another future research avenue is exploring the consequences of these STEM misperceptions. We would expect these STEM misperceptions to influence how likely it is that resources are allocated to Asian Americans. For example, given that participants in our studies considered Southeast Asian subgroups to be underrepresented in STEM, will participants be more likely to consider students from these Southeast Asian subgroups deserving of a scholarship meant for students who are underrepresented in STEM? Understanding this process has implications for Asian applicants and the likelihood that these Asian applicants receive funding during their graduate career and obtain tenure-track positions at research institutions.

## Limitations

Despite our efforts in our studies to examine whether certain groups of people would be more accurate in their STEM or framing to increase STEM estimations, we still found that participants were inaccurate about Asian subgroup representation in STEM. However, there are still some limitations in our study design that may have contributed to these inaccuracies. One is that we limited our definition of STEM representation to include Asian individuals who have an advanced STEM degree, which does not include those with a Bachelor's degree in STEM. According to the U.S. Census[46], when Bachelor's degree-holders are included in the percentage, 70% of the Asian individuals in STEM are from the Chinese and Indian subgroups (compared to 82% of advanced STEM degree holders), while 11% are from the Filipino and Vietnamese subgroups (compared to 4%). In supplemental analyses, we compared the current STEM estimates from Studies 1-2 to the STEM percentages that include Bachelor's degree-holders and replicated our results, except for the Chinese subgroup. However, because our participants were asked to consider advanced degree holders, future research should examine if this is indeed the case.

## Conclusions

Identifying where inequalities exist is essential to addressing them. The Asian American category is often perceived as a successful racial group[25–28], which can lead to the neglect of unique histories and experiences within this diverse group. The present research highlights the consequences of overlooking these within-group inequalities, particularly the misperceptions

about Asian subgroup representation in STEM fields. Our findings underscore the critical role of social science in raising awareness of such disparities. Importantly, institutions can begin to rectify this by collecting and presenting data at the Asian subgroup level to help increase awareness. Failing to educate the U.S. population and continuing to ignore these within-group differences perpetuates inequality, resulting in the persistent underrepresentation of Southeast Asian subgroups in STEM.

## Data availability

Data files are available to download on the Open Science Framework (https://osf.io/g7z8x/?view_only=07b45c2d46a84888af3fd210731e6d41).

## Code availability

Data codes are available on the Open Science Framework (https://osf.io/g7z8x/?view_only=07b45c2d46a84888af3fd210731e6d41).

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

## Acknowledgements

Time spent preparing this manuscript was supported by research grant #G-2111-34528 from the Russell Sage Foundation. The funder had no role in study design, data collection and analysis, decision to publish, or preparation of the manuscript.

## Author contributions

A.C.V. and M.W.K. designed the research; A.C.V. collected and analyzed the data; and A.C.V. and M.W.K. wrote the paper.

## Competing interests

The authors declare no competing interests.
