## [Transparent Peer Review file · Communications Psychology]

The Misperception of Asian Subgroup Representation in STEM

Corresponding Author: Dr Aeroelay Vinluan

Version 0:

Decision Letter:

Dear Dr Vinluan,

Thank you for your patience during the peer-review process. Your manuscript titled "Misperceptions of Asian Subgroup Representation in STEM" has now been seen by 3 reviewers, and I include their comments at the end of this message. They find your work of interest but raised some important points. We are interested in the possibility of publishing your study in Communications Psychology, but would like to consider your responses to these concerns and assess a revised manuscript before we make a final decision on publication.

We therefore invite you to revise and resubmit your manuscript, along with a point-by-point response to the reviewers. Please highlight all changes in the manuscript text file.

Editorially, we consider it important that the revised manuscript consider the alternative analysis approach suggested by R1. Please ensure that interpretation of the findings stays close to the results and is well-situated in the literature and theory.

Please ensure you follow our statistical guidelines when reporting statistics (<https://www.nature.com/commpsychol/submit/submission-guidelines#statistical-guidelines>). Please note in particular our requirements for the reporting and interpretation of null-results. Non-significant findings derived from null-hypotheses significance tests should be reported in full, but may not be interpreted. Where you interpret null results, this interpretation must be based on Bayes Factors or equivalence tests.

When choosing a colour scheme for your figures please consider how it will display to users with colour blindness. Please consider distinguishing data series using line patterns rather than colours, or using optimised colour palettes such as those found at <https://doi.org/10.1038/nmeth.1618>

I am attaching an Editorial Requests Table that details critical reporting requirements for the revised manuscript. Please attend to each item and ensure your manuscript is fully compliant. If your revised manuscript is not aligned with these requests on major issues, such as those concerning statistics, it may be returned to you for further revisions without re-review.

Please submit the following items:

- Revised manuscript
- Point-by-point response to the referees' comments
- Cover letter (as a separate document)

- <https://www.nature.com/documents/nr-reporting-summary.pdf>>Nature Research Reporting Summary
- Completed Editorial Request Table (attached).

via this link: Link Redacted .

Additional guidance is available in our style and formatting guide Communications Psychology formatting guide.

Best regards,

Jennifer Bellingtier

Jennifer Bellingtier, PhD
Senior Editor
Communications Psychology

REVIEWER EXPERTISE:

Reviewers 1, 2, 3: social psychology, group stereotypes, identity

REVIEWER REPORTS:

Reviewer #1 (Remarks to the Author):

Thank you for the opportunity to review for Communications Psychology.

I enjoyed reading the manuscript titled "Misperceptions of Asian Subgroup Representation in STEM". The authors presented four studies that examine people's misperception of Asian Americans subgroups in STEM, a couple of potential explanations for why the misperception may occur, and one possible intervention for the downstream consequences of such misperception. The authors were thorough in their methodological and analytical reporting. I especially appreciate the clever way they have measured misperception (by taking the difference score between STEM representation and population representation). Although the paper has strengths, there are some limitations to the current work that warrant additional attention and reframing before publication.

1. One major limitation of the current work is the construct validity of the main variable in the article, misperception of representation. The authors have operationally defined misperception of representation as the difference between participants' estimated percentage and the actual percentage of each subgroup in STEM. I'm not sure that this definition captures "representation". Their latter conceptualization of over- and under-STEM representation relative to participants' population estimates seems to match closer to the perception of representation they have discussed in the introduction. However, suppose we were to look at the paired t-tests between participants' STEM compared to population estimates and compare that to the actual difference between STEM and population percentages for each group. In that case, we can see that participants are fairly accurate in their perceptions of each subgroup's representation. Using the second definition, the only misperception that seems to emerge is that 1) participants from study 2 perceived that Chinese Americans are well-represented when they are overrepresented, 2) participants perceived that Japanese Americans are overrepresented when they are well-represented, and 3) participants perceived that Korean Americans are well-represented when they may be underrepresented. Thus, perhaps a better analysis of perception of representation would be to conduct a one-sample t-test to compare the difference score between participants' estimates to the difference score between the actual percentages.
2. Perhaps, this is why the intervention in Study 4 didn't have a significant main effect. Participants from studies 1-2 correctly

perceive that Filipino and Vietnamese Americans are underrepresented in STEM relative to their population, which is consistent with actual percentages from the Census Bureau. The intervention simply confirms participants' perception; it didn't present any new information to participants.

3. Related to point 1, an additional problem that arises when measuring it using participants' STEM estimates with actual STEM percentages. Can we reasonably assume anyone to know actual percentages of workers in STEM? You found in Study 3 that participants also misperceive White and Black Americans' representation. Thus, this misperception relative to actual percentage (rather than relative to participants' population estimates) is not unique to Asian Americans. This raises questions about the importance of the current work.

4. Related to the first and third point, the pattern of participants' STEM estimates still roughly matches the pattern of the actual STEM percentages. If we were to look at the pattern and not the absolute value, the misperception that occurred is that participants perceived there to be fewer Indian Americans in STEM relative to Chinese Americans in STEM, when in reality, Indian Americans outnumber Chinese Americans in STEM.

5. Still related to the first point, since participants' estimates for each subgroup are dependent on each other, shouldn't there be at least a Bonferroni correction (by adjusting the significant threshold; $.05 / 7 = .0071$) to control for family-wise error for all of the t-tests? Or, the authors should at the very least acknowledge the potential inflated error rate in the limitation.

6. Results for study 3 are missing from the main article, perhaps due to space concerns. But, it would be interesting for readers to know that misperceptions, based on how the authors have defined it, are also present for White and Black Americans.

7. In study 4, since the intervention is specifically about Filipino and Vietnamese Americans' underrepresentation in STEM, why combine the group-specific items with the other items about affirmative action? Why not examine that item separately?

8. Related to point 5, even if participants did misperceive Filipino and Vietnamese Americans' representation in STEM, why should we expect to see participants' support for affirmative action generally when the intervention was group-specific? Wouldn't it be more reasonable to expect support for affirmative action toward just Filipino and Vietnamese Americans?

Overall, the authors were thorough in their reporting, but they should not overstate the existence of misperception of Asian subgroup representation in STEM.

Reviewer #2 (Remarks to the Author):

The paper presents four studies that examined the (mis)perceptions of Asian American representation in STEM. Asian Americans are commonly associated with STEM fields, but this stereotypic association misses the underrepresentation of certain Asian ethnic groups, particularly Southeast Asian Americans like Hmong, Laotian, Filipino, and Vietnamese Americans. There is much to like about this paper. The topic is understudied and deeply impactful. The studies were rigorously conducted and built on one another nicely. The findings were straightforward and compelling. I believe it will make a strong contribution to the field. I have some suggestions that I hope the authors and editor will find useful.

First, I think the first two paragraphs in the Intro should be revised to be stronger and more compelling. As someone in this area of research, I can immediately see the importance of the topic. However, I don't think the current setup is strong enough for a reader who is unfamiliar with the topic. Although the General Discussion pointed out the severe underrepresentation and under-recognition of Southeast Asians in STEM and in society, the Intro does not explicitly state that. I recommend directly mentioning the percentages of STEM representations early on, or even mentioning that college attendance rate differs greatly by ethnic groups (see NY Times link): <https://www.nytimes.com/interactive/2021/08/21/us/asians-census-us.html?referringSource=articleShare>

Second, I think the General Discussion section on South Asian American representation could benefit from additional theorizing. The authors wrote: "it seems likely that Indian Americans are perceived as more typical of the Asian American category in STEM-context compared to unspecified contexts. Future research should consider the role of context in perceptions of Asian typicality." This is a very interesting finding about the role of context in perceived prototypicality but I found myself wanting to know more. The perceived overrepresentation of Indian Americans (relative to perceived population estimates) could stem from a variety of reasons. There is indeed a strong representation of Indian Americans in STEM (and specifically the tech industry), in large part thanks to the H1B visa. South Asians are also represented (stereotypically) in media such as the characters Raj Koothrappali in Big Bang Theory and Dinesh Chughai in Silicon Valley. Tech CEOs such as Sundar Pichai of Google could further cement the image of South Asian representation in STEM. I believe the authors could easily improve this section.

My last comment is fairly minor. I thought the figures were difficult to interpret, particularly as someone with colorblindness. In Figures 1 and 2, the means and actual values were hard to see amidst all the individual data jitters. I would recommend removing the data points or make the jitter even more transparent. I also had difficulty interpreting Figure 3 as I literally could not differentiate the colors of the top four lines in each panel.

Finally, this recent paper may be useful to cite:

Powdthavee, N. (2025). Curb your positive stereotypes: Counter-stereotypical interventions improve attention to inequality and discrimination against Asian Americans. *Personality and Social Psychology Bulletin*.
<https://journals.sagepub.com/doi/full/10.1177/01461672251341608>

Reviewer #3 (Remarks to the Author):

I appreciate the author's efforts to examine perceived within-group disparities among Asian Americans in STEM fields. The studies present sufficient evidence to show that there is a misperception of Asian subgroup representation in STEM. The studies were preregistered, and the necessary information was provided in the main text and supplemental materials. Below, I provide some questions and suggestions for the authors to consider when revising the manuscript:

1. I find it somewhat difficult to understand how this study contributes to the broader theoretical framework. The findings that suggest misperceptions of Asians sub-groups in STEM offer useful insights, but their implications for theory are somewhat limited. The paper could benefit from a deeper discussion of its relevance to theories of stereotypes and cognitive biases.
2. The video-based intervention was interesting. However, the effects appear to be limited to individuals low in SDO. Given that, isn't it too strong to say that "interventions that raise awareness of inequalities within racial groups by presenting data can promote racial equity-enhancing policies" (p. 22, L414-415)? Just to note, this sentence is missing a period at the end, and the paragraph it belongs to could benefit from a larger indent at the beginning.
3. I understand that the data were collected through Prolific. Personally, I would prefer to see more demographic information (e.g., age, gender, racial identity, education, occupation, etc.) included in the main manuscript rather than in supplemental materials, especially given that the present studies focus on perceptions related to these important demographic categories.

Version 1:

Decision Letter:

Dear Dr Vinluan,

Your manuscript titled "The Misperceptions of Asian Subgroup Representation in STEM" has now been seen by our reviewers, whose comments appear below. In light of their advice I am delighted to say that we are happy, in principle, to publish a suitably revised version in *Communications Psychology*.

We therefore invite you to revise your paper one last time to address the remaining concerns of our reviewers and a list of editorial requests. At the same time we ask that you edit your manuscript to comply with our format requirements and to maximise the accessibility and therefore the impact of your work.

EDITORIAL REQUESTS:

SUBMISSION INFORMATION:

OPEN ACCESS:

* DATA AVAILABILITY:

Link Redacted

Best regards,

Jennifer Bellingtier

Jennifer Bellingtier, PhD
Senior Editor
Communications Psychology

REVIEWER EXPERTISE:

Reviewers 1, 2, 3: social psychology, group stereotypes, identity

REVIEWERS' COMMENTS:

Reviewer #1 (Remarks to the Author):

Thank you for addressing all my concerns in your revised manuscript. You have made a great point about interpretational

challenges with multiple difference scores (Response 1.2) and I agree with your rebuttal. I look forward to reading your manuscript in its published form.

Reviewer #2 (Remarks to the Author):

I was Reviewer #2 in the initial submission. I have read the revised manuscript and the response letter. I am satisfied with the revision and have no further comments.

Reviewer #3 (Remarks to the Author):

I acknowledge the points discussed in the General Discussion (p.24), particularly the authors' reflections on the implications of the study. I understood that the findings contribute to raising awareness of the importance of addressing the needs of the entire category rather than focusing only on typical or default subgroups.

Regarding the second point, I recognize that the disaggregated data, even when independent of SDO, provide stronger empirical support for the effect of video-based intervention. Thank you very much for this clarification.

Finally, I would like to thank the authors for including Table 1, which clearly presents the demographic information and enhances the transparency of the sample characteristics.

October 10, 2025

Dear Reviewers,

Thank you for your helpful comments on our manuscript, “The Misperceptions of Asian Subgroup Representation in STEM.” We hope that we have addressed your comments in our revised manuscript. Below is the original letter with your comments and, in bold, our responses to address each comment.

REVIEWER EXPERTISE:

Reviewers 1, 2, 3: social psychology, group stereotypes, identity

REVIEWER REPORTS:

Reviewer #1 (Remarks to the Author):

Thank you for the opportunity to review for Communications Psychology.

I enjoyed reading the manuscript titled “Misperceptions of Asian Subgroup Representation in STEM”. The authors presented four studies that examine people’s misperception of Asian Americans subgroups in STEM, a couple of potential explanations for why the misperception may occur, and one possible intervention for the downstream consequences of such misperception. The authors were thorough in their methodological and analytical reporting. I especially appreciate the clever way they have measured misperception (by taking the difference score between STEM representation and population representation). Although the paper has strengths, there are some limitations to the current work that warrant additional attention and reframing before publication.

Response 1.1: Thank you for your kind words. We hope that we addressed your comments.

1. One major limitation of the current work is the construct validity of the main variable in the article, misperception of representation. The authors have operationally defined misperception of representation as the difference between participants’ estimated percentage and the actual percentage of each subgroup in STEM. I’m not sure that this definition captures “representation”. Their latter conceptualization of over- and under-STEM representation relative to participants’ population estimates seems to match closer to the perception of representation they have discussed in the introduction. However, suppose we were to look at the paired t-tests between participants’ STEM compared to population estimates and compare that to the actual difference between STEM and population percentages for each group. In that case, we can see that participants are fairly accurate in their perceptions of each subgroup’s representation. Using the second definition, the only misperception that seems to emerge is that 1) participants from study 2 perceived that Chinese Americans are well-represented when they are overrepresented, 2) participants perceived that Japanese Americans are overrepresented when they are well-represented, and 3) participants perceived that Korean Americans are well-represented when they may be underrepresented. Thus, perhaps a better analysis of perception of representation would be to conduct a one-sample t-test to compare the difference score between participants’ estimates to the difference score between the actual percentages.

Response 1.2: Your comment that you are not sure that the difference between the perception of STEM representation and the actual population in STEM captures representation is a good point. You argue further that the difference between STEM perceptions and population perceptions is more appropriate (i.e., “closer to the perception of representation”). We agree with this assessment, and in the paper, you will see how we have amended our analyses and writing in several ways to be consistent with this operational definition of representation.

- **We discuss why the difference between STEM estimates and population estimates is a better metric of representation in the introduction because it accounts for participant theories about subgroup population size (p.4 and p.15).**
- **Though we still compare perceptions of STEM to population size in our first analysis (these analyses remain important to determine if people are accurate or not about STEM estimates), inferences we make about representation judgements in STEM are conducted using the analysis of representation using the operational definition above (p.14-18).**
- **We examine associations between subgroup prototypicality and status this operational definition of representation (p.18-19), finding patterns that are consistent with our prior version of the manuscript.**

You also suggested a couple of additional analyses around other permutations of STEM v. population estimates. One version compares the difference between STEM vs. Population estimates to the difference between STEM vs. Population actual percentages. We do not favor this analysis over the one we report in the manuscript because there are interpretation challenges around where the perception errors come from (e.g., misperception of population, of STEM, of both). For this first analysis (comparing the difference between STEM vs. Population estimates to the difference between STEM vs. Population actual percentages), a meta-analysis of the results across Studies 1 and 2 showed that the estimate differences were less than the actual differences for the Chinese, $Z_{Fisher} = -0.37, p < .0001, 95\% CI [-0.44, -0.31]$, and Indian subgroups, $Z_{Fisher} = -0.55, p < .0001, 95\% CI [-0.62, -0.49]$. However, the estimate difference was greater than the actual difference for the Japanese, $Z_{Fisher} = 0.32, p < .0001, 95\% CI [0.26, 0.38]$, Korean, $Z_{Fisher} = 0.38, p < .0001, 95\% CI [0.27, 0.50]$, Filipino, $Z_{Fisher} = 0.43, p < .0001, 95\% CI [0.34, 0.52]$, and the Vietnamese subgroups, $Z_{Fisher} = 0.12, p = .0012, 95\% CI [0.05, 0.19]$. These analyses indicate that perceptual errors were larger than population differences for everyone but the Chinese and Indian groups, though as we mention, interpreting drivers of these errors is impossible given the multiple difference scores.

The second version of this analysis compares the difference between STEM vs. Population estimates to the STEM actual percentages. For this second version of the requested analysis (comparing the difference between STEM vs. Population estimates to the STEM actual percentages), the estimate differences were less than the actual STEM percentages for the Chinese, $Z_{Fisher} = -1.19, p < .0001, 95\% CI [-1.25, -1.13]$, Korean, $Z_{Fisher} = -0.23, p < .0001, 95\% CI [-0.30, -0.16]$, Indian, $Z_{Fisher} = -1.49, p < .0001, 95\% CI [-1.55, -1.42]$, Filipino, $Z_{Fisher} = -0.43, p < .0001, 95\% CI [-0.51, -0.35]$, and Vietnamese subgroups, $Z_{Fisher} = -0.34, p < .0001, 95\% CI [-0.40, -0.27]$. For the Japanese subgroup, the estimate difference was not significantly different from the actual STEM percentage, $Z_{Fisher} = -0.01, p < .0001, 95\% CI [-0.07, 0.05]$. In general, this pattern indicates that errors were usually larger

between the difference score and STEM actuals, but again, interpretation is compromised by multiple difference scores.

We have chosen not to report these two additional analyses in the manuscript because they include some interpretation challenges around what drives errors in representation (population or STEM errors or both). We hope that overall, including the sharper operational definition of representation and overall streamlining of the results around this definition, has improved the manuscript.

2. Perhaps, this is why the intervention in Study 4 didn't have a significant main effect. Participants from studies 1-2 correctly perceive that Filipino and Vietnamese Americans are underrepresented in STEM relative to their population, which is consistent with actual percentages from the Census Bureau. The intervention simply confirms participants' perception; it didn't present any new information to participants.

Response 1.3: While most people would assume that presenting participants with data or numbers will be sufficient in addressing misperceptions of increasing support, research has shown that this does not always work. Instead, what matters is the framing used when informing participants of statistics. For example, Callaghan and colleagues (2021) found that an intervention with a narrative framing (i.e., highlighting the challenges that Black people face every day) is more effective in addressing misperceptions of the Black-White racial wealth gap over time compared to an intervention with statistics about the Black-White racial wealth gap. Additionally, Hetey & Eberhardt (2014) demonstrate that depending on the framing of the data (i.e., more vs. less) results in differences in policy support. Therefore, we designed our informational intervention to focus on the framing of the data. Specifically, in addition to telling participants that Southeast Asian Americans are underrepresented in STEM, we discuss why underrepresentation is a problem and that policies like affirmative action, which are meant for underrepresented students in STEM do not include Southeast Asian Americans.

In the revised manuscript, we clarify this by first removing that the informational intervention "presented data about the actual Asian subgroup representation in STEM" and adding that the informational intervention "focused on the consequences for people not knowing that these two Asian subgroups are underrepresented in STEM" on p.20. We also added an additional sentence after describing the contents of the informational intervention that "policies like affirmative action do not target Southeast Asian Americans in STEM" on p.20.

3. Related to point 1, an additional problem that arises when measuring it using participants' STEM estimates with actual STEM percentages. Can we reasonably assume anyone to know actual percentages of workers in STEM? You found in Study 3 that participants also misperceive White and Black Americans' representation. Thus, this misperception relative to actual percentage (rather than relative to participants' population estimates) is not unique to Asian Americans. This raises questions about the importance of the current work.

Response 1.4: Please see Response 1.2

Yes, our findings do indicate that participants are likely to misestimate the number of White and Black Americans who have an advanced STEM degree. However, finding this effect among White and Black Americans does not invalidate or make it less unique for

Asian Americans. We believe that the reasons why participants misestimate White and Black Americans in STEM (i.e., belief in racial progress) are likely to be different for why participants misestimate Asian subgroup representation in STEM (i.e., typicality and status perceptions). While important to understand, the representation of White and Black Americans in STEM is not the focus of this research. Our manuscript is attempting to demonstrate the importance of knowing that there is diversity in the Asian American experience, especially since model minority assumptions (e.g., all Asian Americans are academically successful) ignore Asian subgroups who are not as academically successful, such as Filipino and Vietnamese Americans.

4. Related to the first and third point, the pattern of participants' STEM estimates still roughly matches the pattern of the actual STEM percentages. If we were to look at the pattern and not the absolute value, the misperception that occurred is that participants perceived there to be fewer Indian Americans in STEM relative to Chinese Americans in STEM, when in reality, Indian Americans outnumber Chinese Americans in STEM.

Response 1.5: Please see Response 1.2

We still think it is important to know whether participants are accurate in their estimations. However, based on your previous comment, we agree that understanding misperceptions of STEM representation as defined by participants' STEM estimates relative to their population estimates gives us a better understanding of how participants are thinking about STEM representation (i.e., whether a subgroup is overrepresented or underrepresented in STEM) and how those perceptions are related to perceived Asian typicality and status.

5. Still related to the first point, since participants' estimates for each subgroup are dependent on each other, shouldn't there be at least a Bonferroni correction (by adjusting the significant threshold; $.05 / 7 = .0071$) to control for family-wise error for all of the t-tests? Or, the authors should at the very least acknowledge the potential inflated error rate in the limitation.

Response 1.6: We applied a Bonferroni correction to the results in Table 2 (formerly Table 1) on p.12. The only result that changed was for the paired samples t-test comparing participants' STEM estimate to their population estimates for the Japanese subgroup in Study 2. With the Bonferroni correction, the t-value is no longer significant.

6. Results for study 3 are missing from the main article, perhaps due to space concerns. But, it would be interesting for readers to know that misperceptions, based on how the authors have defined it, are also present for White and Black Americans.

Response 1.7: We added a footnote on p.14 to describe the results of Study 3 for White and Black Americans.

7. In study 4, since the intervention is specifically about Filipino and Vietnamese Americans' underrepresentation in STEM, why combine the group-specific items with the other items about affirmative action? Why not examine that item separately?

Response 1.8: In the revised manuscript, we analyzed each affirmative action-related item separately and reported the results for the affirmative action item related to Filipino and Vietnamese Americans (p.20-22). Specifically, "An independent sample t-test revealed that there was not an overall significant difference in participants' reported

support for affirmative action to include Southeast Asian Americans between the intervention ($M=3.28$, $SD=0.92$) and control conditions ($M=3.23$, $SD=0.87$), $t(706)=0.76$, $p=.449$, $d=0.057$. As an exploratory analysis, we examined the interaction between intervention conditions and social dominance orientation (SDO), or support for social inequality and hierarchy, on affirmative action support. Regression results show that SDO significantly and negatively predicts affirmative action support, $b=-0.20$, $p<.0001$, $95\%CI[-0.27, -0.13]$, and when SDO is included in the regression model, affirmative action support is significantly higher in the intervention than in the control condition, $b=0.51$, $p=.0003$, $95\%CI[0.23, 0.78]$, which is consistent with our hypothesis. Furthermore, SDO significantly moderates the relationship between intervention condition and affirmative action support, $b=-0.17$, $p=.0005$, $95\%CI [-0.27, -0.08]$. We followed up the significant interaction with a simple slopes analysis: for participants low in SDO ($-1SD=1.27$), participants in the intervention condition reported greater affirmative action support than participants in the control condition, $b=0.29$, $p<.0001$, $95\%CI [0.12, 0.46]$. However, there were no reported differences in affirmative action support for participants at the mean level of SDO ($M=2.53$), $b=0.07$, $p=.240$, $95\%CI [-0.05, 0.19]$, and those high in SDO ($+1SD=3.79$), $b=-0.14$, $p=.100$, $95\%CI [-0.32, 0.03]$. The SDO moderation analysis replicates previous research findings that participants high in SDO are less likely to support hierarchy-attenuating policies like affirmative action.

8. Related to point 5, even if participants did misperceive Filipino and Vietnamese Americans' representation in STEM, why should we expect to see participants' support for affirmative action generally when the intervention was group-specific? Wouldn't it be more reasonable to expect support for affirmative action toward just Filipino and Vietnamese Americans?

Response 1.9: We still include the results for the other two affirmative action items in our revised manuscripts (p.20-21). Specifically, we were curious if learning about the underrepresentation of Southeast Asian Americans in STEM increased participants' support for affirmative action in general. Perhaps learning about one group that was underrepresented in STEM would lead participants to think about other groups that are underrepresented in STEM as well as in other work or educational domains (i.e., women, Black Americans), leading to greater support for affirmative action overall. However, we did not find any significant differences between the intervention and control conditions for these two general affirmative action items, even when we included SDO as a potential moderator.

Overall, the authors were thorough in their reporting, but they should not overstate the existence of misperception of Asian subgroup representation in STEM.

Response 1.10: Thank you for this comment. We have edited our language throughout our revised manuscript to more closely reflect our findings.

Reviewer #2 (Remarks to the Author):

The paper presents four studies that examined the (mis)perceptions of Asian American representation in STEM. Asian Americans are commonly associated with STEM fields, but this stereotypic association misses the underrepresentation of certain Asian ethnic groups, particularly Southeast Asian Americans like Hmong, Laotian, Filipino, and Vietnamese Americans. There is much to like about this paper. The topic is understudied and deeply impactful. The studies were rigorously conducted and built on one another nicely. The findings

were straightforward and compelling. I believe it will make a strong contribution to the field. I have some suggestions that I hope the authors and editor will find useful.

Response 2.1: We appreciate that you think that this paper will make a strong contribution to the field. We hope that we addressed your comments.

First, I think the first two paragraphs in the Intro should be revised to be stronger and more compelling. As someone in this area of research, I can immediately see the importance of the topic. However, I don't think the current setup is strong enough for a reader who is unfamiliar with the topic. Although the General Discussion pointed out the severe underrepresentation and under-recognition of Southeast Asians in STEM and in society, the Intro does not explicitly state that. I recommend directly mentioning the percentages of STEM representations early on, or even mentioning that college attendance rate differs greatly by ethnic groups (see NY Times link): <https://www.nytimes.com/interactive/2021/08/21/us/asians-census-us.html?referringSource=articleShare>

Response 2.2: In the Introduction (p.2), we added to our discussion of how data on Asian Americans is rarely disaggregated by ethnic subgroup, especially in educational contexts. We then included statistics from the New York Times article as well as from NPR to highlight the underrepresentation of Southeast Asians in college degrees. Specifically, “In 2020, only 33% of Vietnamese Americans were college graduates, compared with 60% of Chinese Americans and 79% of Indian Americans (Gebeloff et al., 2021). Similarly, when looking at graduate degrees, only 10% of Filipino Americans and 10% of Vietnamese Americans had earned one, compared with 29% of Chinese Americans and 43% of Indian Americans (Shivaram, 2021). Despite narratives and stereotypes that Asian Americans are educated and do well in school, these data show variation, with greater representation among Chinese and Indian Americans, and less representation among Filipino and Vietnamese Americans. Given the lack of information about Asian subgroups, many people are unlikely to understand cultural, ethnic, and representational variation within the Asian American category.”

Second, I think the General Discussion section on South Asian American representation could benefit from additional theorizing. The authors wrote: “it seems likely that Indian Americans are perceived as more typical of the Asian American category in STEM-context compared to unspecified contexts. Future research should consider the role of context in perceptions of Asian typicality.” This is a very interesting finding about the role of context in perceived prototypicality but I found myself wanting to know more. The perceived overrepresentation of Indian Americans (relative to perceived population estimates) could stem from a variety of reasons. There is indeed a strong representation of Indian Americans in STEM (and specifically the tech industry), in large part thanks to the H1B visa. South Asians are also represented (stereotypically) in media such as the characters Raj Koothrappali in Big Bang Theory and Dinesh Chugtai in Silicon Valley. Tech CEOs such as Sundar Pichai of Google could further cement the image of South Asian representation in STEM. I believe the authors could easily improve this section.

Response 2.3: We have included further discussion of why there may be a strong association between Indian Americans and STEM in the General Discussion (p.23-24) based on your comments. Specifically, “For example, the H-1B visa program awards 73% of the visas to people from India, and the most common H-1B occupations are computer-related jobs (Pew Research Center, 2025). Additionally, the CEOs of major technology companies such as Alphabet Inc. (Google’s parent company), Microsoft, Adobe, and IBM are of South Asian descent (Fortune, 2025).”

My last comment is fairly minor. I thought the figures were difficult to interpret, particularly as someone with colorblindness. In Figures 1 and 2, the means and actual values were hard to see amidst all the individual data jitters. I would recommend removing the data points or make the jitter even more transparent. I also had difficulty interpreting Figure 3 as I literally could not differentiate the colors of the top four lines in each panel.

Response 2.4: We apologize that we did not consider how our figures may or may not be viewed clearly by individuals who are colorblind. We have addressed this by changing our color palettes to be suitable for colorblindness of various degrees. Additionally, for Figure 1 (formerly Figure 2), we have considered your suggestion by making the data points more transparent. Furthermore, we decided to move Figure 3 to the supplement but have separated the regression lines by Asian subgroup.

Finally, this recent paper may be useful to cite:

Powdthavee, N. (2025). Curb your positive stereotypes: Counter-stereotypical interventions improve attention to inequality and discrimination against Asian Americans. *Personality and Social Psychology Bulletin*.

<https://journals.sagepub.com/doi/full/10.1177/01461672251341608>

Response 2.5: Thank you for suggesting this paper. We found this paper to be very relevant to our manuscript and have cited it on p.6.

Reviewer #3 (Remarks to the Author):

I appreciate the author's efforts to examine perceived within-group disparities among Asian Americans in STEM fields. The studies present sufficient evidence to show that there is a misperception of Asian subgroup representation in STEM. The studies were preregistered, and the necessary information was provided in the main text and supplemental materials. Below, I provide some questions and suggestions for the authors to consider when revising the manuscript:

Response 3.1: We appreciate your comments and hope that we have addressed them.

1. I find it somewhat difficult to understand how this study contributes to the broader theoretical framework. The findings that suggest misperceptions of Asians sub-groups in STEM offer useful insights, but their implications for theory are somewhat limited. The paper could benefit from a deeper discussion of its relevance to theories of stereotypes and cognitive biases.

Response 3.2: Thank you for this suggestion. We added a paragraph in the General Discussion on p.24 that discusses how our results are relevant to cultural defaults (Smith & Zarate, 1992), a form of cognitive biases. Specifically, we discuss that diverse social categories, like Asian Americans, are susceptible to cultural defaults, where typical category members (e.g., East Asian Americans) are automatically thought of when considering the needs of the overall category. Our findings do provide support for this, given that participants were unaware of the extent to less typical Asian subgroups were underrepresented in STEM.

2. The video-based intervention was interesting. However, the effects appear to be limited to individuals low in SDO. Given that, isn't it too strong to say that "interventions that raise

awareness of inequalities within racial groups by presenting data can promote racial equity-enhancing policies” (p. 22, L414-415)? Just to note, this sentence is missing a period at the end, and the paragraph it belongs to could benefit from a larger indent at the beginning.

Response 3.3: You are correct that the effect is limited to individuals low in SDO, but specifically for affirmative action support. We also found a significant difference between video conditions for disaggregated data support, regardless of SDO (p.21-22), and considered collecting and presenting disaggregated data as an additional example of racial equity-enhancing policies. Regardless, we decided to omit this sentence as the sentence prior more accurately summarizes our results.

3. I understand that the data were collected through Prolific. Personally, I would prefer to see more demographic information (e.g., age, gender, racial identity, education, occupation, etc.) included in the main manuscript rather than in supplemental materials, especially given that the present studies focus on perceptions related to these important demographic categories.

Response 3.4: Thank you for this suggestion. We have moved the demographics table from the Supplement to the main manuscript, and it is now Table 1 (p.9).

November 20, 2025

Dear Reviewers,

Thank you for your helpful comments on our manuscript, “The Misperception of Asian Subgroup Representation in STEM.” Below is the original letter with your comments and, in bold, our responses to address each comment.

REVIEWER EXPERTISE:

Reviewers 1, 2, 3: social psychology, group stereotypes, identity

REVIEWERS' COMMENTS:

Reviewer #1 (Remarks to the Author):

Thank you for addressing all my concerns in your revised manuscript. You have made a great point about interpretational challenges with multiple difference scores (Response 1.2) and I agree with your rebuttal. I look forward to reading your manuscript in its published form.

Response 1.1: We're glad that we were able to address your concerns in our revised manuscript.

Reviewer #2 (Remarks to the Author):

I was Reviewer #2 in the initial submission. I have read the revised manuscript and the response letter. I am satisfied with the revision and have no further comments.

Response 2.1: We're glad that you are satisfied with our revisions.

Reviewer #3 (Remarks to the Author):

I acknowledge the points discussed in the General Discussion (p.24), particularly the authors' reflections on the implications of the study. I understood that the findings contribute to raising awareness of the importance of addressing the needs of the entire category rather than focusing only on typical or default subgroups.

Response 3.1: We're glad that we made the study implications clear in the General Discussion. Thank you for this recommendation.

Regarding the second point, I recognize that the disaggregated data, even when independent of SDO, provide stronger empirical support for the effect of video-based intervention. Thank you very much for this clarification.

Response 3.2: We're glad that we can clarify our findings for Study 4.

Finally, I would like to thank the authors for including Table 1, which clearly presents the demographic information and enhances the transparency of the sample characteristics.

Response 3.3: Thank you for suggesting moving Table 1 to the main manuscript.